Optimization of anther culture of awnless triticale

Ma Jun
Zhao Fangyuan
Tian Xinhui
Du Wenhua duwh@gsau.edu.cn
Collage of Pratacultural Science, Gansu Agricultural University , Lanzhou , Gansu , China
Daehler Curtis
Electronic publication date: 2025 Sep 30
Publication date: 2025
Volume: 13
Electronic Location ID: e19951
Received 2024 Dec 18; Accepted 2025 Jul 29
Copyright: ©2025 Ma et al.
Copyright year: 2025
Copyright holder: Ma et al.
License: This is an open access article distributed under the terms of the Creative Commons Attribution License, which permits unrestricted use, distribution, reproduction and adaptation in any medium and for any purpose provided that it is properly attributed. For attribution, the original author(s), title, publication source (PeerJ) and either DOI or URL of the article must be cited.
License URL: https://creativecommons.org/licenses/by/4.0/

Keywords: Triticale, Anther culture, Callus, Doubled haploid, Flow cytometry, Agronomic traits

Funding: The National Natural Science Foundation 32260339 Industry Supporting Program 2022CYZC-49 Key Projects of Gansu Province 21ZD4NA012 Major Science and Technology project of Tibet China XZ202101ZD003N Major Project of Agricultural Biological Breeding 2023ZD0402605-02 This study was supported by the National Natural Science Foundation (32260339), Industry Supporting Program (2022CYZC-49) and Key Projects (21ZD4NA012) of Gansu Province, Major Science and Technology project of Tibet (XZ202101ZD003N), China, and Major Project of Agricultural Biological Breeding (2023ZD0402605-02). The funders had no role in study design, data collection and analysis, decision to publish, or preparation of the manuscript.

==============================
Background

Compared with traditional breeding methods, anther culture is an effective method for quickly obtaining homozygotes within one generation. The method of cultivating doubled haploid plants derived from the anthers of awnless triticale was studied and optimized.

Methods

Young awnless triticale spikes were pretreated at 4 °C for 5, 10, 15, 20, or 25 days, respectively, and then the anthers of different treatment days were cultured on cysteine heart blood (CHB) with antibiotics media with four different hormone concentrations, respectively.

Results

Overall, 15 days was the best low-temperature treatment, and CHB medium containing 1.5 mg/L 2,4-Dichlorophenoxyacetic acid (2,4-D) and 1.5 mg/L Kinetin (KT) was the best hormone concentration treatment. The callus induction rate (CIR) was highest (22.20%) for anthers pretreated for 15-days and inoculated on CHB medium containing 1.5 mg/L 2,4-D and 1.5 mg/L KT. The green plantlet differentiation frequency (DFG) was highest (30.20%) for anthers pretreated for 25-days and inoculated on CHB medium containing 0.5 mg/L 2,4-D+0.5 mg/L KT. Green plantlet production (PRG) was highest (4.58%) for anthers which were pretreated for 10-days and inoculated on CHB medium containing 0.5 mg/L 2,4-D+0.5 mg/L KT. The success rate of chromosome doubling for regenerated green plantlets was 52.8%. Nine of the thirteen DH1 plants (the first generation of double haploid plants) had tip and side awns shorter than 5 mm, implying that they may be used for cultivating awnless triticale.

Conclusion

Triticale anther culture technology was optimized in this work, enabling the rapid breeding of homozygous varieties of awnless triticale and accelerating the breeding of new varieties of awnless triticale.

Introduction

Triticale (×Triticosecale Wittmack) was the first artificial species created by intergeneric hybridization between wheat (Triticum spp.) and rye (Secale cereale) (Evtushenko et al., 2019). As a multipurpose grain-forage species, triticale is an important forage crop (Ashkvari et al., 2024). Triticale is widely cultivated in the alpine pastoral area of northwestern China as a forage crop for feeding livestock (Ma et al., 2023). Most triticale varieties produce awns. If awned varieties are not cut in time, the awns gradually become waxy and cause cuts in the mouths of livestock, thereby increasing the likelihood of pharyngitis, mouth ulceration, and submandibular edema during feeding (Rebetzke, Bonnett & Reynolds, 2016). Awns can also adversely affect the production of high-quality hay (Graves & Ivey, 2018). Therefore, the breeding of awnless triticale varieties is necessary for improving the palatability of the forage crop and increasing the production of high-quality hay. Breeding studies of triticale can improve the hay quality and palatability of triticale, and reduce the damage to the oral cavity of livestock. In addition to traditional breeding methods, anther culture is an indispensable and rapid alternative technique. It has been used to quickly generate haploid and inbred lines of hybrid varieties (Ferreres et al., 2019). More specifically, it can shorten the time required for breeding new varieties by 3∼5 years while also decreasing the amount of human labor and financial resources necessary for breeding (Kostylev et al., 2023). For this technique, the anthers of the F1 or F2 heterozygous generation are cultured in vitro. The microspores in the anthers are induced to dedifferentiate to form a callus, which is subsequently induced to redifferentiate to form a haploid plant (Orlowska et al., 2023). After chromosome doubling, doubled haploid plants are formed, ultimately resulting in the breeding of new or improved varieties (Yuan et al., 2015; Wuerschum et al., 2015).

The efficiency of triticale anther culture is affected by many factors, including the temperature and duration of anther pretreatment as well as the hormone content, plant genotype, pollen grain growth period, and medium type (Pachota, Orlowska & Bednarek, 2022). The anther pretreatment temperature and duration and the hormone content are the most common factors influencing anther culture (Popova et al., 2016). Pretreatment at high or low temperatures can increase the callus or embryoid body induction rate (Lazaridou et al., 2016). This pretreatment can significantly increase the rice, wheat, and pepper anther callus induction rates (Gao et al., 2024; Abe et al., 2020). Low-temperature pretreatment can cause developing microspores to deviate from the gametophyte developmental pathway to the sporophyte developmental pathway, leading to the production of haploid calli or embryoid bodies, which subsequently develop into plants (Orowska et al., 2019). The signal induced by exposure to low temperatures initiates a change in microspore development, which is important for embryoid body formation (Zur et al., 2021). In a previous study, cold treatment of the anthers of 10 winter triticale varieties at 4 °C for 2 weeks significantly increased the callus induction rate, and when the cold treatment was extended to 3 weeks, the ratio of green plants to albino plants of seven varieties increased. The ratio of green plants of the ‘Asmus’ triticale variety increased the most, by 8.3%, and cold treatment had the greatest effect (Immonen & Robinson, 2000). Lusarkiewicz-Jarzina et al. (2017) reported that the triticale anther culture effect was the best after cold treatment for 6 days at 4 °C: the number of winter triticale ‘CT14259’ calli reached the highest level of 118.0 per 100 anthers, and the callus induction effect was the best. Therefore, many laboratories have adopted the method of low-temperature pretreatment in the anther culture of triticale. However, some scholars believe that low-temperature pretreatment is not necessary to obtain high anther culture efficiency. Xynias et al. (2001) found that low-temperature pretreatment had no significant effect on the anther culture response of three wheat varieties (Acheloos, Vergina, and Acheloos×Vergina). Cold pretreatment also often negatively affects the androgenogenic ability of wheat genotypes, thereby reducing embryo or callus production (Ghaemi, Sarrafi & Alibert, 1994). Therefore, it is necessary to explore the effect of low-temperature pretreatment on the anther culture effect of triticale.

Hormones are indispensable in culture media for the formation of calli or embryoid bodies (Zhang et al., 2023). Amp (2003) reported that the addition of 2.0 mg/L 2,4-d and 0.5 mg/L KT to C17 medium increased the callus number of 12 F1 hybrid triticale, and the ‘Ticino × Pronto’ triticale line induced the greatest number of calli, i.e., 107.3 calli per 100 anthers, whereas Zur et al. (2015) reported that adding NAA and KT to the medium (0.5 and 0.5 mg L−2, respectively) also increased the triticale embryoid body induction rate. Kruppa et al. (2023) reported that the combination of 2,4-D and KT was a potent hormonal combination that induced callus formation in triticale anthers culture, and the combination of 1.5 mg/L 2,4-D and 0.5 mg/L KT showed highest callus induction rate and green plantlet differentiation rate. There were differences in the effects of different hormone types and concentrations in the culture medium on the anther culture effect of triticale. Therefore, in order to improve the callus induction rate, it is necessary to further optimize the hormone types and concentrations in the medium.

Chromosome doubling with colchicine solution can transform induced haploid green plants into doubled haploid plants (Touchell, Palmer & Ranney, 2020); however, many researchers have not performed ploidy identification of the induced green plants in time to ensure that the green plants were haploid before chromosome doubling, creating a gap in knowledge. In the process of anther culture, in addition to pollen microspores that induce calli and then differentiate into green plants, the cells of the anther wall, anther septum and filaments may also differentiate to form calli and then differentiate into green plants. The green plants induced by pollen microspores were haploid (triploid in this experiment), whereas the green plants induced by the cells of the anther wall, anther septum, and filament were diploid (hexaploid in this experiment) (Zhang, 2020). Therefore, green plants induced by calli may either be haploid or exhibit other levels of ploidy. In our study, ploidy analysis of the induced green plants was carried out via flow cytometry to determine the ploidy of the regenerated green plants before chromosome doubling with colchicine solution to overcome this shortcoming. In triticale anther culture, the experimental materials used by many researchers are awned triticale genotypes; our experimental material was a new line of awnless triticale selected by our team, and no reports on anther culture of awnless triticale genotypes have been published. In many triticale anther culture experiments, researchers have obtained only doubled haploid green plants but have paid little attention to the agronomic traits of DH0 plants (the plants developed from anther callus) and DH1 plants (the first generation of double haploid plants, which grew from the seeds of DH0 plants) in the field. The phenotypic traits of regenerated plants, significant differences in the same traits of regenerated plants, and whether regenerated plants meet the breeding goals expected by researchers have rarely been reported. In addition, the agronomic traits of the DH0 and DH1 generation of regenerated plants have mostly been reported in other crops, such as wheat and oat (Li et al., 2003; Ponitka & Slusarkiewicz-Jarzina, 2009; Kutlu et al., 2019), but not in triticale. Therefore, we specifically studied the agronomic traits of the DH1 generation of triticale to ensure the integrity and novelty of the experiment. We hypothesized that modifying the low-temperature pretreatment duration and hormone concentration may affect the triticale anther callus induction rate, which can affect the production of haploid plants. After haploid green plants were identified via flow cytometry, chromosome doubling treatment with colchicine was performed to obtain doubled haploids. Therefore, this study provides a reference for the optimization of anther culture technology for awnless triticale.

Materials and Methods

The triticale material used in this experiment was awnless triticale line T2020-6337 (hexaploid, 2n=6x=42) derived from a cross between the male parent T17 (2013) (triticale line) and the female parent J6 (2013) (triticale line). After a 30-day vernalization, the awnless triticale seeds were sown in a plastic greenhouse at Gansu Agricultural University (36°03′N, 103°53′E; 1,560 m above sea level), China, in July 2020. A line seeding method was adopted, with a row spacing of 20 cm and a sowing depth of 3∼4 cm. Additionally, 300 kg hm−2 diammonium phosphate was applied before sowing and 196 kg hm−2 urea was applied (top dressing) at the seeding and jointing stages. Plants were irrigated as required.

‘Portions of this text were previously published as part of a preprint (https://www.researchsquare.com/article/rs-4608942/v1)’.

Sampling and low-temperature pretreatment of young spikes

The samples were collected starting in late October and early November 2020. The young spikes of triticale plants were cut when the pollen grains were in the uninucleated microspore stage (i.e., spikes just reached the middle of the flag leaf and the second leaf from the top). The young spikes were wrapped in aluminum foil, which was labeled with the collection date, and placed in a large beaker containing tap water. Then, the beaker was placed into a bucket with a lid and several ice packs and transported to the laboratory.

Two to four anthers collected per young spike were crushed on a glass slide, after which 1∼2 drops of acetic acid magenta dye (Solarbio Inc., Beijing, China) were added, and the stained material was examined via a Panthera U microscope (eyepiece 10 × objective 40 ×; Motic China Group Co., Ltd., Hong Kong, China) to determine the pollen developmental stage. The young spikes that passed the microscopic examination were equally divided into five groups (each group had a total of no less than 15 young spikes) and then placed separately in five beakers (500 mL beaker filled with 200 mL tap water) for pretreatment at 4 °C for 5 days (A1), 10 days (A2), 15 days (A3), 20 days (A4), or 25 days (A5).

Isolation and culturing of anther

Upon completion of the low-temperature pretreatment, the flag leaves and the second leaf from the top were removed. The surface of the material was disinfected via immersion in 75% ethanol for 1 min in a biological safety cabinet (Sujingantai Inc., Suzhou, Jiangsu, China). Young spikes were obtained, disinfected for 7 min in a 2% sodium hypochlorite solution, and rinsed with sterile water 4∼5 times, and any surface moisture of young spikes was removed using sterile filter paper (Xinhua, Inc., Hangzhou, Zhejiang, China). The anthers were removed from the young spikes via sterile forceps and placed on cysteine heart blood (CHB) induction medium with four different hormone concentrations (denoted as B1, B2, B3, B4, respectively), and 30∼50 anthers were inoculated into a petri dish (90 mm diameter) (Jingan, Inc., Shanghai, China).

In this experiment, four different hormone concentrations were developed by modifying the hormone contents of CHB induction medium. The four different hormone concentrations of CHB induction medium were shown in Table 1.

Table 1 Hormone concentration of induction medium.

Number	Hormone concentration	
B1	0.5 mg/L 2,4-D+0.5 mg/L KT	
B2	1.0 mg/L 2,4-D+1.0 mg/L KT	
B3	1.5 mg/L 2,4-D+1.5 mg/L KT	
B4	2.0 mg/L 2,4-D+2.0 mg/L KT	

Each treatment was completed using eight petri dishes, which were placed in an HGZ-250 incubator (Yuejin Inc., Shanghai, China) for a 3-day incubation at 32 °C in darkness. The petri dishes were transferred to another incubator for a 60-day incubation at 28 °C in darkness. A few calli were observable on Day 30 of incubation. On Day 60, the number of calli was recorded.

Regeneration

Calli with a diameter exceeding one mm were transferred to Murashige and Skoog (MS) differentiation medium (4.74 g/L MS medium + 1 mg/L indoleacetic acid (IAA) + 1 mg/L N6-benzyladenine (6-BA) + 10 g/L sorbitol + 30 g/L sucrose + 7 g/L agar, with a pH of 5.8). The calli were cultured in an incubator set at 27 °C with a 14-h light (2,000∼3,000 lx): 10-h dark cycle. The medium was changed every 15 days. After 60 days, the numbers of green plantlets and albino plantlets were recorded.

Ploidy level analysis of the regenerated green plantlets

Flow cytometry was used to determine the ploidy levels of the regenerated plants. Briefly, 1∼2 leaves were collected from each plant and then placed in 400 µL extraction buffer (CyStain UV Precise P Kit; Sysmex Co., Hamburg, Germany). The leaves were minced for 1 min using a sharp blade and passed through a 30 µm filter to remove cell debris. Next, 1,600 µL of DAPI staining solution was added to stain the nuclei prior to analysis via a CyFlow ploidy analyzer (CyFlow Cube 6; Sysmex Co., Hamburg, Germany). More than 3,000 nuclei were detected per sample. The hexaploid triticale variety ‘Gannong No. 3’ (CK2, provided by Gansu Agricultural University, Lanzhou, Gansu, China) was used as a control to determine whether the regenerated plants were haploid plants.

Chromosome doubling, seedling training, and transplanting

The green plantlets with more than four leaves were transplanted into plastic pots (Jiesheng Co., Shenzhen, Guangdong, China) containing nutrient soil (Luneng Co., Tianzhu, Gansu, China). The pots were incubated in a climatron set at 25 °C with a 14-h light:10-h dark cycle. When the green plantlets had three tillers (or were approximately 12 cm tall), they were removed from the plastic pot, rinsed, and then their roots were soaked in a solution containing 0.1% colchicine, 2% dimethyl sulfoxide, and 0.05% Tween-20 for 5.0 h chromosome doubling step. The roots were rinsed by tap water and then the green plantlets were replanted in plastic pots and incubated in the climatron for 3∼5 weeks to resume growth. They were subsequently transferred to a field.

Ploidy level analysis

Flow cytometry was used to determine the ploidy levels of 36 regenerated plants after they grew normally to the booting stage. The analysis method was the same as ‘Ploidy Level Analysis of the regeneration green plantlets’. Octoploid triticale variety ‘Jinsong No. 49’ (CK3, provided by the Hebei Academy of Agriculture and Forestry Science, Shijiazhuang, Hebei, China) was used as a control to determine whether the regenerated plants were DH plants.

Agronomic trait analysis

The following traits of the regenerated plants were examined at the flowering stage: plant height (distance from the ground to the top of the spike, but excluding the awn, which was measured using a ruler), and number of effective tillers (branch length was more than 50 cm). The following spike parameters were examined at the dough stage: tip awn length (length of the longest awn at the top of the spike, which was measured using a vernier caliper), side awn length (length of the longest awn of the middle spikelet on both sides of the spike, which was measured and then the average value was calculated), spike length (length from the basal spikelet to the tip of the spikelet, but excluding the awn), number of spikelets (number of fertile and sterile spikelets), number of grains per spike, and grain weight per spike (determined using an electronic balance).

Statistical analysis

Each anther-inoculated petri dish was used as a replicate, with no fewer than eight replicates per treatment. The number of anthers used for the inoculation, calli, green plantlets, and albino plantlets in each petri dish was recorded. Additionally, callus induction rate (CIR), green plantlet differentiation frequency (DFG), albino plantlet differentiation frequency (DFA), green plantlet production (PRG), albino plantlet production (PRA), and plant regeneration rate (PRR) were calculated using the following formulae:

CIR (%) = Number of calli/Number of anthers used for the inoculation ×100

DFG (%) = Number of green plantlets/Number of calli ×100

DFA (%) = Number of albino plantlets/Number of calli ×100

PRG (%) = Number of green plantlets/Number of anthers used for the inoculation ×100

PRA (%) = Number of albino plantlets/Number of anthers used for the inoculation ×100

PRR (%) = Total number of plantlets/Number of anthers used for the inoculation ×100.

The differences in CIR, DFG, DFA, PRG, PRA, and PRR were analyzed using the SPSS 20.0 software. In this experiment, Skewness–Kurtosis Test were used to test the data distribution of DFG, DFA, PRG, PRA, and PRR, and —Z-score— > 1.96, the sample data were not normally distributed, therefore, the Kruskal–Wallis test was used in this experiment, and then the Games-Howell post-hoc test was used, and P < 0.05 was statistically significant.

Results

Microscopic examination of anthers

Microscopic examination of the collected young spikes of triticale plants revealed that most of the microspores were in the late uninucleate stage and that the nuclei of individual microspores were located at or near the center, which was ideal for anther cultivation (Figs. 1A, 1B).

Analysis of variance

The results of the analysis of the differences in the CIR, DFR, DFA, PRG, PRA, and PRR are presented in Table 2. There were very significant differences at the 0.01 level in the CIR for the low-temperature pretreatment duration, hormone concentrations, and their interactions. There were also very significant differences at the 0.01 level in the CIR, DFG, DFA, PRG, PRA, and PRR for the interaction between low-temperature pretreatment duration and hormone concentration (Table 2).

Effects of low-temperature pretreatment duration, hormone concentration and their interaction on the average CIR

Low-temperature pretreatment

As the duration of low-temperature pretreatment increased, the average CIR of the triticale anthers initially increased but then decreased. The average CIR for the different hormone concentrations was highest (15.83%) for the 15-day low-temperature pretreatment, followed by the 10-day pretreatment and then the 20-day, 25-day, and 5-day pretreatments. The average CIR was similar for the 10-day and 20-day pretreatments, ranging from 11.94% to 12.92%, which was higher than the average CIR for the 5-day pretreatment (<10%). Accordingly, the average CIR was highest for the anthers pretreated at a low temperature for 15 days, reflecting the suitability of this pretreatment for anther culture (Fig. 2).

Figure 1 Micrographs of triticale microspores at late uninucleate stage.

(A) Microspores of T2020-6337 line (1,000× magnification); (B) the magnified late uninucleate microspore of T2020-6337 line (B was a partially enlarged area of A) Note: the arrow pointed to the location of the nucleus of the microspore.

Table 2 Kruskal–Wallis test results of CIR, DFG, DFA, PRG, PRA, and PRR in different treatment.

Variation	Index	Chi-Squareχ (χ2)	Degree of freedom (df)	P value (P)	
-Low temperature pretreatment days	CIR (%)	16.837	4	0.002**	
Hormone concentrations	CIR (%)	8.896	3	0.031**	
Low temperature pretreatment days × hormone concentration	CIR (%)	48.145	19	0.000**	
DFG (%)	72.109	19	0.000**	
DFA (%)	37.151	19	0.008**	
PRG (%)	65.587	19	0.000**	
PRA (%)	36.060	19	0.010**	
PRR (%)	76.083	19	0.000**	
Notes.

** Indicates significant differences at the 0.01 level.

CIR callus induction rate

DFG green plantlet differentiation frequency

DFA albino plantlet differentiation frequency

PRG green plantlet production

PRA albino plantlet production

PRR plant regeneration rate

The same as below.

Figure 2 Effects of the low- temperature pretreatment duration on the average CIR for different hormone concentrations.

CIR, callus induction rate. The Skewness–Kurtosis test and the Games–Howell post-hoc test was used in this experiment, and different letters above the columns indicate significant differences at the 0.05 level.

Hormone concentration

Increases in hormone concentrations were accompanied by a decrease in the average CIR of triticale anthers pretreated at 4 °C for different durations. The average CIR of B3 was greater than those of B1 and B2, but it was not significantly different from that of B1 and B2. The average CIR of B4 was significantly lower than those of B1, B2, and B3. Thus, B3 had the highest average CIR, implying that it may be best for anther culture (Fig. 3).

Figure 3 Effects of hormone concentrations on the average CIR of triticale for different low-temperature pretreatment durations.

CIR, callus induction rate; B1, CHB induction medium with 0.5 mg/L 2,4-D + 0.5 mg/L KT; B2, CHB induction medium with 1.0 mg/L 2,4-D + 1.0 mg/L KT; B3, CHB induction medium with 1.5 mg/L 2,4-D + 1.5 mg/L KT; B4, CHB induction medium with 2.0 mg/L 2,4-D + 2.0 mg/L KT. The Skewness–Kurtosis test and the Games–Howell post-hoc test was used in this experiment, and different letters above the columns indicate significant differences at the 0.05 level.

Interaction between these factors

Analysis of the interaction between low-temperature pretreatment duration and hormone concentration indicated that for the same low-temperature pretreatment duration, for A1, the CIR was greater or significantly greater for B1 than for the other media. For A3, the CIR was greater or significantly greater for B3 than for the other media. For A4 and A5, the CIR was greater or significantly greater for B2 than for the other media. Thus, the hormone concentrations needed for a high CIR varied depending on the duration of the low-temperature pretreatment. For the same hormone concentration, for both B1 and B3, the CIR was highest for A3 and then A2, with lower or significantly lower CIRs for A5, A1, and A4. For B2, the CIR was greater or significantly greater for A4 than for the other low-temperature pretreatment durations. For B4, the CIR was highest for A2. Hence, the duration of low-temperature pretreatment required for a high CIR differed among the tested hormone concentrations. Overall, the CIR was significantly greater for A3B3 (22.20%) than for the other treatment combinations, with the exception of A3B1 (Fig. 4).

Figure 4 Effects of the interaction between the low-temperature pretreatment duration and hormone concentration on CIR.

CIR, Callus induction rate; B1, CHB induction medium with 0.5 mg/L 2,4-D + 0.5 mg/L KT; B2, CHB induction medium with 1.0 mg/L 2,4-D + 1.0 mg/L KT; B3, CHB induction medium with 1.5 mg/L 2,4-D + 1.5 mg/L KT; B4, CHB induction medium with 2.0 mg/L 2,4-D + 2.0 mg/L KT. The Skewness–Kurtosis test and the Games–Howell post-hoc test was used in this experiment, and different letters above the columns indicate significant differences at the 0.05 level.

Effects of the interaction between low-temperature pretreatment duration and hormone concentration on DFG and DFA

DFG

For A1, DFG was significantly greater for B2 than for the other media (Fig. 5A), whereas for A2, A4, and A5, DFG was significantly greater for B1 than for the other media. For A3, DFG was significantly greater for B4 than for the other media. Therefore, the hormone concentrations needed for a high DFG varied depending on the duration of low-temperature pretreatment. For the same hormone concentrations, for B1, DFG was significantly greater for A5 and A4 than for A1, A2, and A3. For B2, the DFG was significantly greater for A1 than for A2, A3, A4, and A5. For B3, the DFG was highest for A5. For B4, DFG was highest for A3. Thus, the duration of the low-temperature pretreatment required to maximize DFG differed among the hormone concentrations. Overall, DFG was significantly greater for A5B1 (30.20%) than for the other treatment combinations (Fig. 5A).

Figure 5 Effects of the interaction between the low-temperature pretreatment duration and hormone concentration on (A) green plantlet differentiation frequency (DFG) and (B) albino plantlet differentiation frequency (DFA).

DFG, green plantlet differentiation frequency; DFA, albino plantlet differentiation frequency; B1, CHB induction medium with 0.5 mg/L 2,4-D + 0.5 mg/L KT; B2, CHB induction medium with 1.0 mg/L 2,4-D + 1.0 mg/L KT; B3, CHB induction medium with 1.5 mg/L 2,4-D + 1.5 mg/L KT; B4, CHB induction medium with 2.0 mg/L 2,4-D + 2.0 mg/L KT. The Skewness–Kurtosis test and the Games–Howell post-hoc test was used in this experiment, and different letters above the columns indicate significant differences at the 0.05 level.

DFA

For the same low-temperature pretreatment duration, for A1 and A4, DFA was greater or significantly greater for B3 than for B2 and the other media (Fig. 5B). For A2, the DFA of B2 was the highest. For A3, the DFA of B4 was the highest, and for A5, the DFA of B1 was the highest, which was significantly greater for the other media. For the same hormone concentration, the DFA of A5 was highest in B1, the DFA of A1 was highest in B2 and B3, and the DFA of A3 was highest in B4, and these values were significantly greater than those of the other media. The A1B4, A2B3, A3B1, A4B4, and A5B4 treatment combinations resulted in the lowest DFA (0.00%; not presented in Fig. 5B), which was suitable for anther culture. These results reflected the variability in DFA among the hormone concentrations and low-temperature pretreatment durations (Fig. 5B).

Differences in green plantlet production, albino plantlet production and plant regeneration rate

PRG

For the same pretreatment day of A2, A3, A4 and A5, PRG was highest for B1. Moreover, there was no significant difference between A3 and A5 in terms of PRG. For A1, the highest PRG (for B2) was significantly higher than the PRG for B1 and B4. For the same hormone concentration treatments, PRG was highest for A2 in B1. For B2, the highest PRG (for A5) was significantly higher than the PRG for A1, A2, A3, A4. For B3, PRG was highest for A3, but it did not differ significantly from that of A5. For B4, the highest PRG (for A3) was significantly higher than the PRG for A1, A2, A4 and A5. These findings indicated that the duration of the low-temperature pretreatment and the hormone concentrations significantly affected PRG (Fig. 6A).

Figure 6 The difference of green plantlet production (PRG) (A) albino plantlet production (PRA) (B) and plant regeneration rate (PRR) (C) in treatments.

PRG, green plantlet production; PRA, albino plantlet production; PRR, plant regeneration rate; B1, CHB induction medium with 0.5 mg/L 2,4-D + 0.5 mg/L KT; B2, CHB induction medium with 1.0 mg/L 2,4-D + 1.0 mg/L KT; B3, CHB induction medium with 1.5 mg/L 2,4-D + 1.5 mg/L KT; B4, CHB induction medium with 2.0 mg/L 2,4-D + 2.0 mg/L KT. The Skewness–Kurtosis test and the Games–Howell post-hoc test was used in this experiment, and different letters above the columns indicate significant differences at the 0.05 level.

PRA

For the same pretreatment day treatment of A1, there were no significant differences in PRA among B1, B2, and B3. However, for A2, A3 and A4, PRA was higher or significantly higher for B2 than for the other media. For A5, PRA was highest for B1. For the same hormone concentration treatments, PRA was highest for A4 in B2 and B3. Additionally, for B1, PRA was significantly higher for A5 than for A1, A2 and A4. For B4, PRA was significantly higher for A3 than for A2. The A1B4, A2B3, A3B1, A4B4, and A5B4 treatment combinations had the lowest PRA (0.00%; not presented in Fig. 7B), indicative of the significant effects of the duration of the low-temperature pretreatment and the hormone concentrations on PRA (Fig. 6B).

Figure 7 Results of DNA content in regenerated seedlings of triticale determined by flow cytometry.

(A) ‘Gannong No. 3’ triticale (CK); (B) haploid regenerated seedlings.

PRR

For the same pretreatment day treatment of the A2, A3, A4 and A5, PRR was higher or significantly higher for B1 than for the other media. For A1, the highest PRR (for B2) was significantly higher than the PRR for B1, B3 and B4. For the same hormone concentration treatments, the analysis of the effects of the different hormone concentrations indicated that PRR was relatively high and consistent for B1 and B2. For B1, PRR was highest (5.59%) for A2, followed by A5 (5.30%) and A4 (4.34%); these values were significantly higher than the corresponding value for A1. For B2, PRR was highest (4.99%) for A5, and for B3, PRR was highest (4.00%) for A3. For B4, PRR was significantly higher for A3 (3.50%) than for A1, A3, A4, and A5 (Fig. 6C).

Determination of the ploidy levels of the regenerated plants

During anther culture, in addition to pollen microspores, the cells of the anther wall, anther septum, and filament may also dedifferentiate to form calli and then differentiate into green seedlings. The green seedlings induced by pollen microspores were haploid (triploid in this test), whereas the green seedlings induced by the cells of the anther wall, anther septum, and filament were diploidized (hexaploid in this test); thus, ploidy analysis of the induced green seedlings was carried out via flow cytometry to determine whether the regenerated green seedlings were haploid. The triticale material (T2020-6337) used in this experiment was hexaploid (2n=6x=42), and the control triticale ‘Gannong No. 3’ (CK2) was also hexaploid (2n=6x=42). If the fluorescence intensity of the relative nuclear DNA content of the regenerated triticale seedlings peaked at 1/2 of that of the control group, the seedlings were considered haploid; otherwise, they were considered not haploid. The test results revealed that the position of the peak of the ‘Gannong No. 3’ triticale control (CK2) was 24,000 (Fig. 7A), while the position of the peak of the regenerated plants was 12,000, with the peak appearing at 1/2 the value of the control indicating haploidy (Fig. 7B). A total of 52 haploid plants were obtained via ploidy analysis of the regenerated plants that survived after training (Table 3).

Chromosome doubling and ploidy identification of the regenerated seedlings after chromosome doubling

After chromosome doubling of haploid plantlets with colchicine solution, 43 plantlets survived; the survival rate was 82.7%. The 43 plantlets were transplanted to the field, and 36 green plantlets survived and adapted to the growth environment of the field (Table 3). Flow cytometry was used to determine the ploidy of the 36 green plantlets (before the booting stage) that grew normally after transplanting in the field. The control used in this test was the octoploid triticale ‘Jinsong 49’ (CK3). If the tested triticale peak was 3/4 of the control (CK3), the tested plants were hexaploid (doubled haploid); that is, chromosome doubling was successful. Figure 8A shows the results of flow cytometry analysis of ‘Jinsong 49’ triticale (CK3). The peak of ‘Jinsong 49’ (CK3) ranged from 48,000∼60,000, while the peak of the tested plants ranged from 32,000∼40,000, which was 3/4 that of the control (Fig. 8B) and indicating hexaploidy (doubled haploid). Figure 8C shows that the test plants peaked at 16,000–28,000, 42,000–48,000 and 48,000–56,000, so they were mixoploid. In the test results, there was a great difference in the high and low peaks, which was related to the amount of sample but not to plant ploidy.

Table 3 Changes in the number of regenerated plantlet before and after chromosome doubling.

Name	TNGP	NHGP	NSGPCD	SR%	NGPT	NSGP	DH	MOPP	SRCD/%	
T2020-7337	57	52	43	82.7	43	36	21	14	58.3	
Notes.

TNGP the total number of green plantlet induced

NHGP the number of haploid green plantlet

NSGPCD the number of survival green plantlet after chromosome doubling

SR survival rate

NGPT the number of green plantlet transplanted in field

NSGP the number of survival green plantlet

DH double haploid plant

MOPP mixoploidy and other ploidy plants

SRCD the success rate of chromosome doubling

Figure 8 Results of DNA content in regenerated seedlings of triticale determined by flow cytometry.

(A) Octoploid triticale variety ‘Jinsong No. 49’ (CK); (B) double haploid regenerated seedlings; (C) mixoploid regenerated seedlings.

The ploidy analysis of the 36 plants that grew normally to the booting stage in the field revealed that 21 plants were doubled haploid (Table 3, Fig. 8B), and the rest were mixoploid or of other degrees of ploidy (Table 3, Fig. 8C); the success rate of chromosome doubling was 58.3% (Table 3).

Agronomic trait analysis

The calli that formed in anther culture differentiated to form haploid green plants, which were transplanted into the field after chromosome doubling. The agronomic traits of the doubled haploid regenerated plants that grew to the flowering stage in the field were measured (Fig. 9). Since DH0 generation plants were strongly affected by tissue culture, the traits of DH0 generation plants were significantly different from those of CK plants. To reduce the influence of tissue culture on plants, the traits of the DH1 generation population were measured in this experiment. There were 19 doubled haploids in the DH0 generation, but six plants were not seeded, so the DH1 generation consisted of only 13 plants. The tip and side awn lengths were less than five mm for plants Z1, Z2, Z7, Z8, Z9, Z10, Z11, Z12 and Z13 (Table 4, Fig. 10), which satisfies the threshold for awnless triticale. Hence, these nine plants may be used as preparatory materials for the cultivation of awnless triticale varieties. The remaining plants had tip and side awn lengths of 5∼10 mm, making them useful preparatory materials for the cultivation of short-awn triticale varieties. The height of the 13 plants ranged from 106.3∼156.3 cm, which was close to that of mature triticale varieties. Except for Z1, Z2 and Z12, the plant height was significantly lower than that of the CK plants. The effective tiller number of the 13 plants was between 3.7 and 5.7, and there was no significant difference between the other plants and the CK, except for Z2 and Z9. In terms of spike traits (especially in number of grains per spike and grain weight per spike), Z12 presented the best spike traits among the 13 plants, and there were no significant differences between Z12 and CK in terms of spike length, spikelet number, number of grains per spike, or grain weight per spike. The next was Z2. The spike traits of the remaining 11 plants were lower or significantly lower than those of CK, indicating that further breeding would be needed if the regenerated plants were to be cultivated into varieties (Table 4).

Figure 9 Production of doubled haploid triticale plants.

(A) Collected triticale young spikes. (B) Induction medium inoculated with anthers. (C) Calli. (D) Differentiated green plantlet and albino plantlet. (E) Growing green plantlet. (F) Green plantlets before chromosome doubling. (G) Green plantlets after chromosome doubling in a climatron. (H) Green plantlets in the field.

Table 4 Agronomic traits of 13 DH1 generation plants.

Number	Plant height/cm	Number of effective tiller	Tip awn/mm	Side awn/mm	Spike length/cm	Number of spikelet	Number of grains per spike	Grain weight per spike/g	
CK	150.9 ± 2.5bc	4.3 ± 0.3bc	4.6 ± 0.1b	4.3 ± 0.2b	13.7 ± 0.8ab	33.0 ± 1.5ab	46.1 ± 1.5ab	2.30 ± 0.1a	
Z1	147.8 ± 0.9c	4.3 ± 0.3bc	4.0 ± 2.0b	4.2 ± 1.0b	12.4 ± 0.2c	30.7 ± 1.3bcd	37.2 ± 1.5cd	1.48 ± 0.1de	
Z2	153.8 ± 0.8ab	5.7 ± 0.4a	4.1 ± 0.3b	3.9 ± 0.4b	13.6 ± 0.3ab	33.3 ± 0.7ab	41.0 ± 1.2bc	2.03 ± 0.3b	
Z3	114.6 ± 2.5 h	3.7 ± 0.3c	8.5 ± 0.7a	7.7 ± 0.7a	12.9 ± 0.3bc	29.3 ± 0.6cde	47.3 ± 1.5a	1.74 ± 0.5c	
Z4	121.4 ± 1.2 g	4.7 ± 0.4abc	8.5 ± 0.4a	7.9 ± 0.8a	12.7 ± 0.2bc	30.7 ± 0.7bcd	49.0 ± 1.7a	1.95 ± 0.1b	
Z5	127.1 ± 1.0f	4.7 ± 0.3abc	7.9 ± 0.6a	7.0 ± 0.3a	12.9 ± 0.3bc	32.7 ± 0.3ab	49.3 ± 2.2a	1.63 ± 0.6cd	
Z6	132.4 ± 1.1e	5.3 ± 0.4ab	8.6 ± 0.7a	8.0 ± 0.8a	14.3 ± 0.2a	34.7 ± 0.7a	49.7 ± 1.5a	2.08 ± 0.1b	
Z7	127.8 ± 0.7f	4.7 ± 0.3abc	4.3 ± 0.2b	2.0 ± 0.5c	12.8 ± 0.4bc	31.0 ± 1.0bc	45.7 ± 2.3ab	1.64 ± 0.4cd	
Z8	110.8 ± 0.9 h	4.3 ± 0.3bc	4.0 ± 0.4b	2.1 ± 0.1c	12.2 ± 0.2c	27.3 ± 0.7e	35.7 ± 1.8cd	1.95 ± 0.5b	
Z9	142.7 ± 1.4d	5.7 ± 0.3a	3.6 ± 1.0b	4.0 ± 0.1b	10.7 ± 0.4d	28.2 ± 1.2de	32.3 ± 1.5de	1.19 ± 0.3f	
Z10	106.3 ± 1.5i	4.7 ± 0.7abc	4.8 ± 0.1b	3.7 ± 0.6bc	12.4 ± 0.3c	28.0 ± 1.2de	33.7 ± 1.9de	1.16 ± 0.2f	
Z11	125.8 ± 1.2f	4.7 ± 0.3abc	3.8 ± 0.6b	2.6 ± 0.5bc	10.6 ± 0.3d	26.7 ± 0.7e	28.7 ± 1.3e	1.40 ± 0.3e	
Z12	156.3 ± 1.3a	5.0 ± 0.6ab	4.5 ± 0.8b	3.0 ± 0.4bc	13.4 ± 0.4abc	33.0 ± 0.6ab	46.0 ± 2.1ab	2.13 ± 0.1ab	
Z13	125.7 ± 0.8f	4.7 ± 0.3abc	4.0 ± 0.2b	3.9 ± 0.1b	12.6 ± 0.3bc	30.7 ± 0.7bcd	46.1 ± 2.6ab	1.96 ± 0.5b	
Notes.

Different lowercase letters in each column indicate significant differences (P < 0.05).

Figure 10 Spike of Z1 triticale plant.

Discussion

Effects of low-temperature pretreatment on the CIR and the differentiation rate of triticale

Among the many factors affecting the CIR of a triticale anther culture, low-temperature pretreatment can lead to a significant increase in the CIR (Garda et al., 2020). Low-temperature pretreatment can induce the development of microspores to deviate from the gametophyte developmental pathway and enter the sporophyte developmental pathway, thereby producing haploid calli or embryoid bodies, which then develop into plants (Lantos et al., 2023). Low-temperature pretreatment can also significantly increase the abscisic acid (ABA) content in anthers, thereby promoting the antioxidant activities of microspores in the early stage of in vitro culture and maintaining microspore viability (Li et al., 2021). Slusarkiewicz-Jarzina et al. (2017) reported that pretreatment at 4 °C for 14 days can significantly increase the production of green winter triticale plants (CT 14259, Mo 35957, Mo 35981, Mo 36082 and Mo 36229) while also improving the androgen response of the plants. With respect to wheat (Sel.9/DH150, Premio/5009, DL41/DH150, DL45/DH150 et al.) anther cultures, some researchers have increased the number of green plants by including a 14-day low-temperature pretreatment (Kanbar et al., 2020), but other researchers have suggested that a 21-day low-temperature pretreatment may be better (Sinha & Eudes, 2015).

Many laboratories now consider low-temperature pretreatment of the young spikes of triticale plants (2∼3 weeks at 4 °C) to be optimal for inducing microspore embryogenesis (Kanbar et al., 2020). In the present study, young triticale spikes were pretreated at 4 °C. The analysis of the effects of pretreatment duration indicated that the CIR was highest (22.20%) for the 15-day low-temperature pretreatment. This result is consistent with the findings of Slusarkiewicz-Jarzina et al. (2017). Moreover, these findings may reflect the ability of the microspores to dedifferentiate during incubation at low temperatures for 15 days. Increasing the duration of low-temperature pretreatment beyond 15 days did not have additional positive effects on the anther culture, which may be indicative of cold-induced damage to the anthers and decreased microspore activity. In our experiment, the A3B3 treatment produced the highest CIR (22.20%), while Gonzalez, Hernádez & Jouve (2010) reported a CIR of 0.73%∼25.0% after 15 days of low-temperature pretreatment of anthers of 13 triticale varieties (except for Presto, CL × To, To × Bo); furthermore, the CIR of the A3B3 treatment in our experiment was significantly greater than that of the 13 triticale varieties reported by Gonzalez, Hernádez & Jouve (2010). Zur et al. (2014) reported the CIR of four ‘recalcitrant’ triticale lines was 2.5%∼22.0%, and the CIR of the A3B3 treatment in our experiment was also higher than that of the four ‘recalcitrant’ triticale lines reported by Zur et al. (2014). These findings indicate that, with the exception of low-temperature pretreatment, this effect is most likely caused by different genotypes.

In this study, the timing of the initial appearance of calli appeared to influence whether green or albino plantlets were ultimately generated. More specifically, the calli that appeared within the first 30 days continued to grow (diameter greater than five mm) and differentiated into green plantlets. In contrast, the calli that appeared after 30 or 60 days grew more poorly (diameter of approximately 1∼2 mm). Additionally, these calli were more likely to develop into brown or albino seedlings during the later induction and differentiation periods, possibly because the calli were too small to differentiate into green shoots (Slusarkiewicz-Jarzina et al., 2017). Hence, even though many calli appeared during the latter part of induction period, these calli produced fewer green seedlings than the calli that appeared earlier.

Albinism, which is a common problem associated with plant tissue culture, is caused by the inability to produce chloroplasts, resulting in a lack of photosynthesis (Migneault et al., 2018). Albinism has been observed in the tissue cultures of monocots, including wheat, barley, and rye. A recent study revealed that prolonged pretreatment often results in a single base pair mutation (C to T), which leads to a missense mutation (Thr to Ile)-related albinism (Pattnaik et al., 2020). The cells of albino seedlings produced via tissue culture may contain totipotent nuclei, but their plastids, which cannot differentiate normally, may be affected by a new sporophyte-related process (Ohnoutkova, Vlcko & Ayalew, 2019). Without chlorophyll, albino plants cannot grow normally in nature, and do not represent any agronomic value. Warzecha et al. (2005) reported that albino plantlets have represented 42.0% of all triticale regenerated plantlets, while in other studies, albino plantlets have outnumbered green regenerants 2.6-ford (Tuvesson, von Post & Ljungberg, 2003). For barley anther and microspore cultures, the proportion of albino plantlets reportedly varies widely (1%∼99.7%) depending on the genotype, with albinism more common in spring barley than in winter barley (Makowska & Oleszczuk, 2014). No correlation was found between green plantlets and albino plantlets formation, suggesting that the cellular mechanisms that control the formation process of green plantlets and albino plantlets were different (Krzewska et al., 2015). In the present study, DFA ranged from 0% to 23.33% among the treatments, with the albino plantlets affecting the plant differentiation efficiency. The production of albino plantlets is related mainly to gene mutations or abnormal gene expression. It is also influenced by culture conditions, including temperature, light, hormones, and mineral elements. Krzewska et al. (2015) used QTL analysis method to find that the production of triticale albino plantlets was related to 14 chromosomal regions among subgenome B (4 QTLs) and R (10 QTLs), and there were 4 QTLs related to albino plantlets formation on the 4R chromosome. Considerable research has been conducted to decrease the DFA. For example, the addition of Cu2+ or Ag+ to medium reportedly alters the green plantlet-to-albino plantlet ratio by decreasing the number of albino plantlets by 18.7% (Warcho et al., 2021). Decreasing the temperature and light intensity may decrease DFA to some extent without affecting normal callus growth and differentiation (Pattnaik et al., 2020). However, the genotype affects the production of albino plantlets considerably more than do culture conditions.

Effects of hormones on the triticale anther CIR

Hormones (type and concentration) have important regulatory effects on cell division, growth, and differentiation. Earlier research on anther cultures indicated that the hormone requirements of different plant species and different genotypes of the same species vary (Lordan et al., 2017). In the induction medium, different hormone types and concentrations affect callus and plantlet formation (Juzoń-Sikora et al., 2022). In anther cultures of many plants, 2,4-D is often used because of its effects on pollen initiation and division as well as the formation of calli and embryoid bodies (Dissanayake et al., 2020). Moreover, KT is one of the most commonly used cytokinins for increasing the efficiency of winter wheat anther cultures (Wang et al., 2023). For triticale anther culture, supplementing the medium with two mg/L 2,4-D reportedly increases the average number of calli (13.9%) (compared with the effects of 6-BA) (Hassawi, Qi & Liang, 2006). In an earlier investigation involving wheat, relatively few calli generated on medium lacking 2,4-D failed to differentiate into plantlets, but increasing the 2,4-D content in the medium from 0.5 mg/L to 1.0 mg/L increased the number of calli and the plantlet regeneration rate (Zheng & Konzak, 1999). However, because prolonging the 2–4 mg/L 2,4-D treatment beyond the initial stage reportedly adversely affects plant regeneration, the 2,4-D concentration in the induction medium should be maintained at 1∼2 mg/L (Zheng & Konzak, 1999). In the present study, pretreatment of triticale anthers at 4 °C for 15 days before the inoculation of medium supplemented with 1.5 mg/L 2,4-D and 1.5 mg/L KT resulted in the highest CIR (22.20%). This result is in accordance with the findings of a previous study by Zheng & Konzak (1999), but this result was higher than the findings of Hassawi, Qi & Liang (2006) (13.9%) and Sun (2009) (16.9%). It indicated that pretreatment of awnless triticale anthers for 15 days before the inoculation of medium supplemented with 1.5 mg/L 2,4-D and 1.5 mg/L KT, the effect of callus induction was the best. For hybrid rice, the highest CIR (21.66%) was obtained when N6 medium containing 2 mg/L 2,4-D and 0.5 mg/L N6-benzyladenine was used, possibly reflecting the synergistic effects of auxin and cytokinin on calli (Pattnaik et al., 2020). Therefore, the synergistic effects of different auxin and cytokinin concentrations on calli should be explored.

The effect of time of colchicine treatment on survival and the success rate of chromosome doubling

A colchicine treatment can increase the probability of regenerating polyploid plants, but it can also cause seedlings to wither and die (Huy et al., 2019). Chromosome doubling agents inhibit spindle fiber formation, thereby preventing the copied chromosomes from separating and migrating to the poles (Broughton et al., 2020). In a previous investigation, the survival rates after the roots and crowns of regenerated oat plants were treated with 0.1% and 0.2% colchicine for 4 h were 97.9% and 93.6%, respectively (Ferrie et al., 2014). In the chromosome doubling study of wheat regenerated plants, the survival rates of the F1 and F2 wheat regenerated plants treated with 0.2% colchicine for 20 min were 72.16% and 91.06% (Kutlu et al., 2019). In another study, the survival rate of whole germinated maize seedlings treated with 0.04% colchicine for 5 h was 47.11%, whereas the survival rates following the treatment of maize seedling roots and crowns with 0.1%, 0.4%, and 0.7% colchicine solutions for 5 h were 88.66%, 94.17%, and 89.00%, respectively (Chaikam et al., 2020). In the present study, the examination of the green triticale plantlets treated with a 0.1% colchicine solution for 5 h revealed that more than half of the plantlets had withered leaf tips after 1 week, while some of the plantlets transferred to plastic pots gradually withered, and the survival rate of green plantlets was 82.7%. When the chromosome doubling time was decreased to 4.5 h, the number of withered plantlets decreased and the survival rate increased, and these findings may reflect the toxicity of colchicine. The results of our study were higher than those of Kutlu et al. (2019) on F1 wheat (72.16%), but lower than those of Ferrie et al. (2014) on oats (93.60%) and Kutlu et al. (2019) on F2 wheat (91.06%), which may be related to regenerated plantlets of different plants or different genotypes of the same plant.

In the DH0 regenerated plants, the plant height, number of grains per spike, and grain weight per spike were generally low (the date was not shown in the paper), and significantly lower than those of the control group (parents, T2020-6337). However, compared with DH0 regenerated plants, the plant height, number of grains per spike, and grain weight per spike of DH1 regenerated plants were significantly increased, and the agronomic traits of DH1 regenerated plants had been basically close to those of the parents (T2020-6337), so the agronomic traits of DH1 regenerated plants were used in this experiment.

The detrimental effects of chromosome doubling chemical agents on plant survival have been reported (Ebrahimzadeh et al., 2018). Therefore, appropriately decreasing the chromosome doubling time can increase the survival rate of regenerated plants. The utility of colchicine has been assessed in terms of chromosome doubling efficiency and plant survival rates, but the potential toxicity of colchicine to plants and animals (including humans) has often been ignored, so researchers have recently started to screen for less toxic alternatives to colchicine (Touchell, Palmer & Ranney, 2020). Amiprophos-methyl and pronamid, which have been used for the chromosome doubling of maize haploid plants, may be better than other alternatives, with an overall success rate (16.1%) that is close to that (22.1%) of colchicine (Melchinger et al., 2015). For regenerating cucumber plants, the highest DH regeneration rates for colchicine and trifluralin are 58.33% and 83.33%, respectively, and the chromosome doubling rates for 750 mg/L colchicine and 50 mL/L trifluralin are reportedly 57.41% and 62.50%, respectively, implying trifluralin may be better than colchicine (Ebrahimzadeh et al., 2018). Although there are viable alternatives, colchicine is still used in many laboratories for the chromosome doubling of haploid plants.

Different colchicine concentrations and treatment times influenced the chromosome doubling rate of the regenerated green seedlings. 215 regenerated haploid triticale plantlets were treated with a 4% colchicine solution for 6 h under light at 25 °C, which resulted in 128 DH plantlets (i.e., chromosome doubling efficiency of 59.9%) (Slusarkiewicz-Jarzina & Ponitka, 2003). The highest haploid transformation rate (80%) for regenerated oat plants was obtained following a 4-h treatment with 0.2% colchicine (Ferrie et al., 2014). Moreover, soaking the roots of maize seedlings in 0.10% and 0.70% colchicine solutions for 5 h can result in chromosome doubling efficiencies of 31.83% and 27.67%, respectively (Chaikam et al., 2020). In our experiment, the roots of the regenerated plants were immersed in a 0.1% colchicine solution for 5 h. Nineteen of the 36 regenerated plants were verified as DH plants, chromosome doubling efficiency was 58.3%, which is similar to the earlier studies by Slusarkiewicz-Jarzina & Ponitka (2003), but higher than the findings of corn studies by Chaikam et al. (31.83%) (2020). It indicated that soaking the triticale roots with 0.1% colchicine solution for 5 h has a better effect. However, it was lower than oats for the highest haploid conversion rate (80.0%), which may be related to different plants. More specifically, clearly, colchicine concentrations and treatment times may need to be adjusted to maximize the chromosome doubling efficiency.

Conclusion

The low-temperature pretreatment days and hormone concentration significantly affected the anther culture success rate of awnless triticale. Among the five low-temperature pretreatment days, 15 days was the best, and among the four hormone concentrations, CHB medium containing 1.5 mg/L 2,4-D and 1.5 mg/L KT was the best. The highest CIR was obtained by pretreating anthers for 15 days at 4 °C prior to inoculation on CHB medium containing 1.5 mg/L 2,4-D and 1.5 mg/L KT. Among the 36 regenerated green plants that were grown in the field, The nine DH1 plants reached the standard of awnless triticale, which could be used as materials for breeding new awnless triticale varieties.

Supplemental Information

Supplemental Information 1 Raw data

Supplemental Information 2 CHB and MS medium formulation

Supplemental Information 3 Highlights

Supplemental Information 4 Codebook

We wish to thank Dr. Nan Xie (Hebei Academy of Agriculture and Forestry Science) and Dr. Xiaohu Lin and Dr. Han Li (Henbei Normal University of Science and Technology) for providing octoploid triticale seeds, and we also appreciate Liwen Bianji (Edanz) and Springer Nature for editing the English text of a draft of this manuscript.

Additional Information and Declarations

Competing Interests

Author Contributions

Data Availability

The authors declare there are no competing interests.

Jun Ma conceived and designed the experiments, performed the experiments, analyzed the data, prepared figures and/or tables, and approved the final draft.

Fangyuan Zhao performed the experiments, analyzed the data, prepared figures and/or tables, and approved the final draft.

Xinhui Tian conceived and designed the experiments, authored or reviewed drafts of the article, and approved the final draft.

Wenhua Du conceived and designed the experiments, authored or reviewed drafts of the article, and approved the final draft.

The following information was supplied regarding data availability:

Raw data is available in the Supplemental Files.

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
