# Peer review of "Optimization of anther culture of awnless triticale"

_PeerJ, doi:10.7717/peerj.19951_

## Round 0.1 · original submission · Minor Revisions

Dear colleagues, your manuscript has now been assessed by experts in the field, and the reviews of your research are completed. I also agree with the reviewers that your work has merit for publication following revision

Reviewer 1 ·

Basic reporting

The abstract is dense, with technical jargon that might hinder accessibility for a broader audience. Simplifying key findings while retaining specificity could improve comprehension.

Certain sections, such as the methods, present information in a convoluted manner. For instance, the description of low-temperature pretreatments and their setup could be restructured for clarity.

Although the literature cited is relevant, some references lack critical discussion regarding contrasting findings in similar studies, which could help contextualize the results further.
Some figures (e.g., Figures 4 and 5) have overlapping data points that reduce interpretability; improved graphical presentation is needed.

Experimental design

The experimental design does not fully explore variability in other factors influencing callus induction and differentiation, such as genotype interaction or environmental conditions during culture.
Statistical analyses are comprehensive but lack justification for the choice of SPSS and Duncan’s test. Alternatives, such as mixed-model ANOVAs, might provide deeper insights into interaction effects.
The authors do not report any control conditions for anthers without pretreatment or hormone treatments, which would strengthen claims of optimization.

Validity of the findings

The discussion lacks a critical assessment of limitations, such as albinism rates and their underlying genetic causes, which could affect generalizability.

Limited information is provided on the physiological or molecular mechanisms driving the observed effects, leaving room for further exploration of hormone interactions.

While the success of A3B3 is clear, the study does not justify why certain hormone concentrations were chosen or how these might vary across other genotypes or species.

Additional comments

Abstract: Simplify the language and structure it into clear subsections (e.g., Background, Objectives, Methods, Results, and Conclusions).
Figures and Tables: Improve the resolution and presentation of data, especially where legends overlap with graphical elements.
Discussion: Include a comparative analysis with findings from other species or genotypes to contextualize the results better.
Controls: Incorporate untreated anthers or hormone-free media as baseline controls in future experiments.
Statistical Rationale: Expand on the rationale behind the chosen statistical methods and explore more sophisticated alternatives if applicable.

Cite this review as

Reviewer 2 ·

Basic reporting

Dear authors
Your manuscript is quite interesting, original and labor intensive. It constitutes an important resource for future anther culture and breeding studies. In general, the language of the manuscript is good, understandable and fluent. The methods were applied correctly. The presentation of the data and the discussion are sufficient. However, a few points mentioned below should be corrected.
-Make the title shorter and more striking. A title like "Optimization of anther culture of awnless triticale" would be better.
Abstract
-Line 18: What does code A3 refer to? You did not specify it before.
- Line 19: What is B1?
- Reorganize the abstract section with a little more detail.
Introduction
- Line 29: Are you sure that the triticale currently used commercially is obtained from Triticum aestivum? Were the hexaploid triticales obtained with Triticum durum more successful? Even if you have produced an octoploid triticale, you should state this when making a general statement.
- Line 37: Here, it would be more appropriate to make a paragraph and add a transition sentence when moving on to the statement about anther culture. For example; "hay quality of triticale can be improved with breeding studies".
- Line 44: It would be more appropriate to use the term "doubled haploid" instead of "diploid". Because what is done with chromosome doubling does not exactly correspond to diploidy.
- Line 46: You can use the term "genotypes" instead of "varieties (lines)".
- Line 45-49: Move these sentences to the end of the introduction as appropriate. You are stating the aim of the study.
- Line 62: It is not "white plants" but "albino plants".
- Line 68: Here, a paragraph should be made when talking about hormones. Do not start with "Although". You can write "Hormones are indispensable in culture media for the formation of calli or embryoid bodies".
- Line 86: It is not "double" but "doubled"
- Line 92: Are you sure it is not?
“Kutlu I., Z. Sirel, O. Yorgancilar, A. Yorgancilar (2019): Line × tester analyses for anther culture response of bread wheat (Triticum aestivum L.- Genetika, Vol 51, No.2, 447-461.”
"Kutlu, I., Çelik, S., Karaduman, Y., & Yorgancılar, Ö. (2023). Phenotypic and genetic diversity of doubled haploid bread wheat population and molecular validation for spike characteristics, end-use quality, and biofortification capacity. PeerJ, 11, e15485."

I recommend you to review these article.

Experimental design

Materials and Methods
- Line 101: What is T17? Wheat? Rye? So what is J6? Write clearly. You should write with the entire audience in mind.
- Line 108: You should provide more detailed information about hormones. This is the subject of your study. Explain
- Line 119: Write clearly how many spikes you used for each application. Were there any replication? Explain
- Line 159: You said that we obtained hexaploid triticale. Why did you use octoploid triticale as a reference?

Validity of the findings

Results
-Line 198: In Table 2, the variance analysis of low temperature treatment and hormone treatment did not show significance at the 1% level. This is probably due to the 0-valued data obtained. Therefore, to increase the reliability of the variance analysis, you can accept 2 petri dishes as 1 replication and eliminate the 0 values. Rearrange the variance analysis.
- Line 275: You said octoploid in Line 159!

Additional comments

Discussion
- Line 344-346: Zur et al. (????). Revise the sentence. There is some confusion.
- Line 411-425: Give examples of wheat as plants closer to triticale.
- Include a brief discussion of the changes in agronomic traits in the DH0 and DH1 generations. You can compare them to other plants, if not triticale, or comment on why the results are the way they are. Discussion is needed.

Cite this review as

---

## Round 0.2 · Minor Revisions

The external reviewers were satisfied by your revision; however, I have identified a number of suggested English edits as well as some concerns regarding your description and presentation of statistics to be addressed as follows:

The abstract (line 24) uses an acronym/abbreviation (“DH1 plants”) that needs to be explained This abbreviation should also be explain when first used in the Introduction (together with DH0).

L 23 “which pretreated” to “which were pretreated”
L 12 “with the anthers” - I suggest “derived from the anthers”
L 17 “15 d was the best low-temperature days” – I suggest “Overall, 15 days was the best low-temperature treatment”

L 39 “The breeding” to “Breeding”
L 198 - This section needs to specify that ANOVA was used (or whatever method you used needs to be stated), not just that Duncan’s multiple comparison test was used,
When you responded to the reviewer’s concern about statistical analysis, you seem to recognize that ANOVA and the Duncan’s multiple comparison test assume that the data are normally distributed, but there is no indication in the manuscript Methods that you checked whether the data were approximately normally distributed. Typically % data are not normally distributed because they are bounded by 0 and 100. For cases when the data are not normal, Kruskal-Wallis may be used followed by the Games-Howell post-hoc test, or the authors could try to transform data to make them closer to normal distributions, then use ANOVA. But this needs to be explained in the Methods.

Table 2 – there is no “PPG” on the table or table Title and there is no “PPA” on the table or on the Table 2 title, but these abbreviations are listed in the note.
Table 2 – The F-values are meaningless unless the numerator and denominator degrees of freedom are provided. These could be given in columns after the “Variation” column and before the “F-value” part of the table. However, this assumes that the % values were roughly normally distributed, which would not be expected – maybe the values were transformed before analysis? This should be explained in Methods. Otherwise, non-parametric statistics like Kruskal-Wallis would be presented in place of F-statistics.

Figure 2 caption – please indicate the type of statistical test used to identify differences.
Figure 3-6 caption – pleas add “Different letters above the columns indicate significant differences at the 0.05 level.” And indicate the type of statistical test used to identify differences.
Figures 2 and 3 – Y-axis label -- change “The average” to “Average”

Figure 3 – “B1”, “B2” etc. are letters and letters are not measured in mg / l. The x-label should specify the actual hormone concentration and the figure caption should specify which hormone is depicted on the x-axis. Or, you could list "B1" B2" etc on the x-axis and then explain what these mean in the figure caption. Then you don't need to add "mg / l" on the x-axis because the letters and not quantities.

Figure 4-6 -- In the figure captions please specify the meaning of B1-B4.

L 477 “effect” change to “success rate”
L 477 “the 15 days” to “15 days” (delete “the”)

Reviewer 1 ·

Basic reporting

The authors have implemented the necessary corrections and revisions, and the manuscript is suitable for acceptance in its current form

Experimental design

The authors have implemented the necessary corrections and revisions, and the manuscript is suitable for acceptance in its current form

Validity of the findings

The authors have implemented the necessary corrections and revisions, and the manuscript is suitable for acceptance in its current form

Additional comments

The authors have implemented the necessary corrections and revisions, and the manuscript is suitable for acceptance in its current form

Cite this review as

Reviewer 2 ·

Basic reporting

I see that the authors have followed all my suggestions to improve the manuscript. From my point of view, everything is OK.

Experimental design

I see that the authors have followed all my suggestions to improve the manuscript. From my point of view, everything is OK.

Validity of the findings

I see that the authors have followed all my suggestions to improve the manuscript. From my point of view, everything is OK.

Additional comments

I see that the authors have followed all my suggestions to improve the manuscript. From my point of view, everything is OK.

Cite this review as

---

## Round 0.3 · accepted · Accept

Previous reviewer comments have been addressed in the revised manuscript
Please make a simple English edit in the Abstract as follows:
L 19 “which pretreated” to “pretreated” (delete "which")